# Differential influence of habit components on compulsive and problematic reward-seeking behavior

**Lavinia Wuensch**[1,2]*, **Yoann Stussi**[1,2], **Théo Vernede**[2], **Ryan J. Murray**[3], **David Sander**[1,2], **Julie Péron**[1,2,3], **Eva R. Pool**[1,2]

**1** Faculty of Psychology and Educational Sciences, University of Geneva, Geneva, Switzerland, **2** Swiss Center for Affective Sciences, University of Geneva, Geneva, Switzerland, **3** Department of Neurology, Cognitive Neurology Unit, University Hospitals of Geneva, Geneva, Switzerland

* lavinia.wuensch@unige.ch

**Data availability statement:** Data and code used to generate the figures and results

## Abstract

Habitual behavior has been identified as a key process involved in a variety of mental health problems. Previous research has shown that habit is not a unitary construct. The present study examined how different components of habitual behavior relate to compulsive and problematic reward-seeking behavior. In two experiments, participants (N = 666) completed a French version of the Creature of Habit Scale, which measures two components of habitual behavior: routine and automaticity. Participants also completed questionnaires assessing compulsivity, problematic reward-seeking behaviors, stress, and impulsivity. Dynamic network analyses indicated that the two habitual behavior components were differentially related to these mental health problems: routine was associated with compulsivity, while automaticity was associated with problematic media consumption. These findings suggest that taking the non-unitary architecture of habit into account may help to better understand the role of habit in mental health.

## Introduction

Habit is a fundamental driver of behavior in everyday life [1–6] and is thought to be a key process involved in various mental health problems [7–9]. Habits are acquired through stimulus-response learning: actions produced in response to a stimulus are assigned value and repeated if they have previously led to a desired outcome. Once learned, the habitual response comes to be reflexively triggered by the stimulus, regardless of the current value of the outcome [1–4]. While this makes habitual behavior efficient, it also introduces a behavioral rigidity that can become maladaptive in dynamic environments. Established habits, being impervious to changes in individuals' goals and motivations, can be very hard to break—even when they result in aversive consequences for the individual [9]. This rigidity has been suggested to play a role in various mental health problems [7–9].

Habits exert control over behavior alongside elaborate goal-directed mechanisms, whereby actions are performed based on a prospective computation of the current outcome value [1,3]. Consequently, goal-directed behavior can flexibly adapt to changing environmental

reported in this manuscript is available at https://github.com/ Affect-Learning-Decisions-Lab/Habit_Affect.

**Funding:** This work was supported by the Swiss National Science Foundation (Ambizione grant PZPGP1 193120 to ERP). YS is supported by an ERC Starting Grant (INFORL-948671) awarded to Prof. Dr. Maël Lebreton. The funders had no role in study design, data collection and analysis, decision to publish, or preparation of the manuscript.

**Competing interests:** The authors have declared that no competing interests exist.

contingencies such as modifications in outcome value. However, this flexibility comes at the expense of a higher computational cost [1–4]. The degree to which individuals rely more on habitual or goal-directed mechanisms is variable, and this variability has been proposed as a transdiagnostic mechanism underlying both compulsive behaviors, such as the repetitive actions or rituals which occur in obsessive-compulsive disorder, and a variety of problematic reward-seeking behaviors [9,10], such as gambling [11], binge eating [12,13], and problematic Internet use [14].

Here, we propose that the heterogeneity of mental health problems associated with habitual behavior could be better understood by taking into account the multidimensionality of habit [15–18]. Indeed, while in the mental health context habit is typically referred to as a unitary construct, models with componential architectures have more recently been suggested [15,19]. These models differentiate a latent habit component corresponding to the "process by which a stimulus generates an impulse to act as a result of a learned stimulus-response association" from habitual behavior, defined as any behavior generated by the latent habit process [15, p. 277]. Habitual behavior can further be separated into initiation and execution components [15]. The conceptualization of habit as a non-unitary construct has led researchers to postulate more complex interactions between goal-directed and habitual control: behavior could be habitual at different levels—habitually initiated, habitually executed, or both—and goal-directed control may be more or less involved in behavior categorized as habitual [15,16]. Such models lead to a departure from dominant competitive accounts of goal-directed and habitual control [3,20], providing a new lens through which links between habits and mental health can be explored.

In recent years, tools have been developed to investigate the non-unitary architecture of habits, among which the Creature of Habit Scale (COHS; [21]). The COHS is a 27-item questionnaire specifically aimed at assessing the tendency toward habits in daily life. Results from the validation of the original COHS, conducted with an English-speaking population, identified two distinct components of habitual behavior: *automatic responses* and *routine behaviors* [21].

Automatic responses refer to relatively simple behaviors (e.g., eating biscuits) that are automatically initiated upon the perception of environmental cues (e.g., a packet of biscuits) and executed until completion (e.g., finishing off the packet). These instances of simple, purely habitual behavior are captured by the COHS automaticity subscale, which measures behavior that is both habitually initiated and habitually executed—that is, completely controlled by habit (e.g., "I often find myself finishing off a packet of biscuits just because it is lying there"). As such, it is closely related to typical experimental operationalizations of habit, where habitual behavior is often measured by cue-triggered, short behavioral responses, such as key-presses or food consumption [20]. By comparison, routines correspond to behaviors that may be habitually initiated or habitually executed—that is, partially under goal-directed control [20]. This includes chunked action sequences executed with a high degree of automaticity [17,21], but dependent on deliberative control to be initiated as components of multistep goal-directed behavioral responses [16,20]. The COHS routine subscale measures individuals' general propensity to show habitual behavior in their daily life (e.g., "I tend to like routine"); that is, a tendency or preference for structured habitual action sequences which may nevertheless be directed toward a goal. For instance, always parking one's car or bicycle in the same place might represent a habitually executed chunk of a goal-directed overarching "going to the movies" behavioral sequence [20,21]. While closer to many real-life occurrences of complex habitual behavior involving a degree of goal-directed control, this dimension of habit is less frequently studied in experimental paradigms [16]. In fact, habitual responding during contingency degradation has been found to correlate with the automaticity subscale, but not

with the routine subscale [22], supporting the idea that the COHS captures a dimension of habitual behavior that is missed by traditional habit tasks. Accordingly, the COHS can be understood to measure a broad range of habitual behaviors that vary in terms of goal-directed involvement.

The COHS's ability to capture various components of habitual behavior makes it particularly relevant for investigating the links between individual differences in habit tendencies and vulnerabilities to various mental health problems. For instance, recent findings show that individuals with cocaine use disorder scored higher than control patients on the automaticity subscale, but not on the routine subscale [22]; similarly, problematic alcohol consumption has been positively linked to automaticity and negatively linked to routine [23]. By contrast, compulsivity appears to be more strongly associated with routine than with automaticity [24,25]. These promising findings highlight the possibility that different components of habit may be differently implicated in mental ill health, with automaticity being involved in problematic reward-seeking behaviors and routine in compulsive behaviors in particular. While there is still debate regarding whether the initiation of compulsive behaviors is automatic, these findings are in line with the growing consensus on the rigid habit-like nature of the execution of compulsive behaviors [26].

In the present study, we examined whether different components of habitual behavior were differently associated with a wide array of problematic reward-seeking and compulsive behaviors. To do so, we adopted a dimensional approach to mental health [27], which conceptualizes mental health problems as dynamic systems interacting within and across diagnostic categories [28]. We used factor analysis to extract mental health dimensions from questionnaires assessing various problematic behaviors. We then implemented network analyses to characterize the links between two components of habitual behavior and dimensions of problematic reward-seeking and compulsive behaviors. Specifically, we tested whether the automaticity component of habitual behavior was associated with reward-seeking behaviors and whether the routine component of habitual behavior was associated with compulsivity. Dimensions which have been shown to modulate the interplay between habitual, reward-seeking, and compulsive behaviors were also included in the network model. We included an affective dimension related to stress, anxiety, and depression, as these are known to exacerbate the symptomatology of many problematic reward-seeking behaviors [10,29,30] and to impact the balance between goal-directed and habitual behavior [31–36]. We also included impulsivity, which has been shown to be positively associated with a variety of problematic reward-seeking behaviors (e.g., compulsive shopping, substance use disorder, binge eating; [37–39]) and automaticity [24], and negatively associated with routine [24,25].

## Materials and methods

### Ethics statement

Experiment 1 was approved by the ethics committee of the Psychology Section of the Faculty of Psychology and Educational Sciences at the University of Geneva (protocol number PSE.20201182.MM). Experiment 2 was approved by the ethics committee of the University of Geneva (protocol number CUREG-2022-02-25). For both experiments, participants gave written informed consent.

### Participants

Following a dimensional perspective, we operated under the assumption that mental health difficulties can be conceptualized as continuous dimensions within the population [27,72].

Consequently, we recruited a non-clinical population in both experiments. There were no inclusion or exclusion criteria based on mental health diagnosis, and participants were not asked to report whether they had received a formal diagnosis or whether they were undergoing treatment for mental health difficulties.

**Experiment 1.** A total of 404 undergraduate psychology students from three different classes ($N_{class\,1}$ = 122; $N_{class\,2}$ = 80; $N_{class\,3}$ = 202) completed the experiment for course credits. The recruitment period ran from April 22, 2020 to November 30, 2021. Participation requirements were a minimum age of 18 years and fluency in French. Three participants were excluded because they only used the extreme values of the questionnaire scales, and 20 additional participants were excluded because they had missing data on questionnaires of interest. The final sample consisted of 381 participants (*mean age* = 22.20 $\pm$ 5.29; 306 women, 72 men, and 3 non-binary). All participants were fluent in French. The sample size was established based on the strategy of collecting as much data as possible from the student pool we had access to during the academic years the experiment was conducted in.

**Experiment 2.** To determine the sample size for Experiment 2, we ran a simulation-based power analysis [40] using an expected network structure derived from Experiment 1. The analysis indicated that a sample of 275 participants was necessary to obtain a correlation of at least 0.8 between the true and the estimated network for edge weights and strength, and for sensitivity. Anticipating for potential dropout, we tested a total of 285 participants (*mean age* = 26.78 $\pm$ 9.83; 188 women, 94 men, and 3 non-binary). We recruited 195 undergraduate psychology students that completed the experiment for course credits. The remaining 90 participants were recruited through the online platform Prolific and received a monetary compensation of 9£ (~11.5$) per hour for their participation. All participants were fluent in French. The recruitment period ran from May 14, 2022 to March 31, 2023.

## Procedures

**Questionnaire translation**  We translated the COHS (27 items; [21]) and the State-Trait Inventory for Cognitive and Somatic Anxiety (STICSA; 42 items; [41]) into French. Both questionnaires were translated by two bilingual French and English speakers. One bilingual speaker first translated each item from English to French, the other bilingual speaker subsequently translated the French items back into English. The back translated versions were compared to the original questionnaires, discrepancies were discussed and translation adjustments were consensually made. Since the COHS subscales are central to the research question of the present study, the validation of the psychometric properties of the French version is presented in S1 File. The French version of the COHS exhibited a structure in two factors: routine (16 items) and automaticity (11 items), replicating the structure of the original [21] and German [25] versions.

**Questionnaire administration**  We selected questionnaires which have been validated in French and which target reward-seeking behaviors, compulsivity, impulsivity, as well as factors related to affective stress. For reward-seeking behaviors, we specifically selected questionnaires targeting behaviors which have shown sufficient variance for the modeling of individual differences in community samples in previous pilot studies (e.g., media consumption, problematic eating). Participants completed the French version of all questionnaires online through the platform LimeSurvey [42].

**Experiment 1.** To assess problematic reward-seeking behaviors, we administered the Eating Attitudes Test 26 (EAT; 26 items; [43]), the Internet Addiction Test (IAT; 20 items; [44]), the Questionnaire About Buying Behavior (QABB; 19 items; [37,45]), and the short version of the Problematic Mobile Phone Use Questionnaire (PMPUQ; 15 items; [46]). To assess

compulsivity, we used the Obsessive-Compulsive Inventory Revised (OCI; 18 items; [47]). We additionally administered the State-Trait Anxiety Inventory (STAI; 40 items; [48]), the Perceived Stress Scale (PSS; 10 items; [49]), the Center for Epidemiologic Studies Depression Scale (CES-D; 20 items; [50]) and the Post-traumatic Checklist Scale (PCLS; 17 items; [51]) to capture mental health problems typically associated with problematic reward-seeking behaviors and compulsivity. Finally, we used the UPPS Impulsive Behavior Scale (UPPS, long version; 40 items for classes 1 and 2; [52]; UPPS-P short version; 20 items for class 3; [53]) to assess impulsivity.

Participants from classes 1 and 2 completed the questionnaires in two separate days (COHS, EAT, IAT, PCLS, PMPUQ, QABB, OCI, and UPPS on day 1 and CESD, PSS, and STAI on day 2). Participants from class 3 completed only a selection of these questionnaires in a single day (CESD, COHS, EAT, IAT, PMPUQ, PSS, OCI, STAI, and UPPS-P). The order of the questionnaires was randomized across participants within each day.

Only questionnaires completed by all participants were included in the analysis (Table 1). We did not include the state portion of the STAI, as only the trait portion reflects characteristics that are stable over time. However, all the data are made openly available (see data and code availability statement). Descriptive statistics for the questionnaires included in the analysis for Experiment 1 are reported in Table 1 and the distributions of the questionnaire scores are shown in Fig 1A. Distributions of the COHS subscales are shown in S1 File.

In order to evaluate the role of habit components in relation to a wider range of problematic reward-seeking behaviors and to estimate whether the relationships we identified in Experiment 1 were replicable using different measures for similar constructs, we ran a second experiment with additional questionnaires.

**Experiment 2.** To assess problematic reward-seeking behaviors, we reused multiple questionnaires from Experiment 1: the EAT [43], IAT [44], and QABB [37,45]. We added the modified Yale Food Addiction Scale 2.0 (mYFAS 2.0; 11 items; [54]) as an additional measure of problematic eating, and replaced the PMPUQ with the newer Smartphone Addiction Scale Short Version (SASSV; 10 items; [55]), which we thought might capture current smartphone use better. In order to explore how habit might interact with additional problematic reward-seeking behaviors, we also administered the Gaming Addiction Scale (GAS; 7 items; [56]),

**Table 1. Cronbach's $\alpha$, mean and standard deviation of the questionnaires used in the analysis for Experiment 1.**

| Questionnaire | N | $\alpha$ | Mean | SD |
|---|---|---|---|---|
| CESD | 381 | 0.93 | 20.85 | 11.94 |
| COHS | 381 | 0.83 | 86.18 | 13.80 |
| EAT | 381 | 0.89 | 9.64 | 10.11 |
| IAT | 381 | 0.90 | 39.03 | 13.39 |
| OCI | 381 | 0.87 | 20.22 | 10.94 |
| PMPUQ | 381 | 0.69 | 2.00 | 0.42 |
| PSS | 381 | 0.87 | 30.53 | 7.09 |
| UPPS/UPPS-P | 263/118 | 0.73/0.74 | 101.30/49.07 | 19.83/7.19 |
| STAI-T | 381 | 0.92 | 47.38 | 10.76 |

$\alpha$ = Chronbach's $\alpha$, SD = standard deviation. CESD = Center for Epidemiologic Studies Depression Scale; COHS = Creature of Habit Scale; EAT = Eating Attitudes Test; IAT = Internet Addiction Test; OCI = Obsessive-Compulsive Inventory Revised; PMPUQ = Problematic Mobile Phone Use Questionnaire-Short Version; PSS = Perceived Stress Scale; STAI-T = State-Trait Anxiety Inventory: trait; UPPS = UPPS Impulsive Behavior Scale; UPPS-P = UPPS-P Impulsive Behavior Scale.
The overall questionnaire score corresponds to the mean for the PMPUQ and to the sum of all items for the remaining questionnaires.

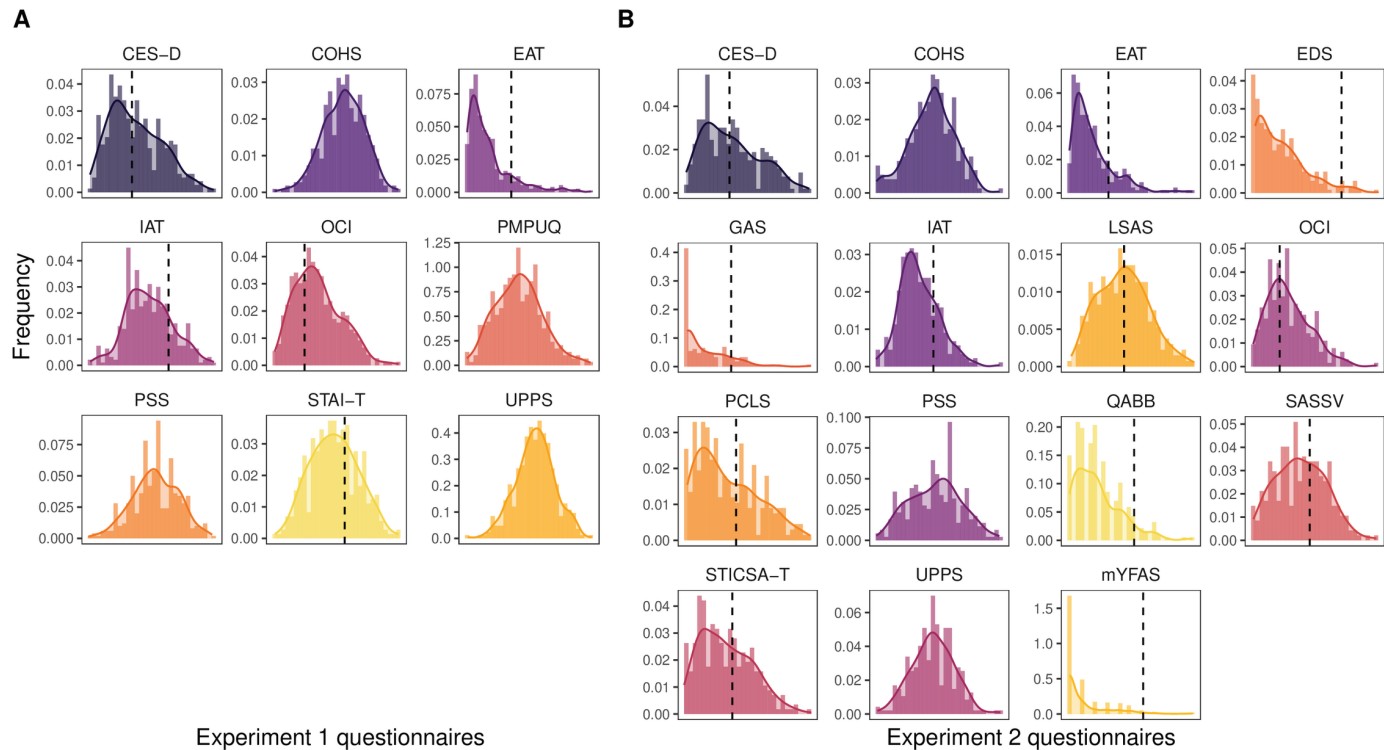

**Fig 1. Questionnaire distributions.** A: Experiment 1. B: Experiment 2. Dashed lines indicate cutoff scores for the CES-D, EAT, IAT, OCI, STAI-T, LSAS, PCLS, QABB, SASSV, and STICSA-T; dashed lines indicate what proportion of the sample is categorized as having problematic behaviors for the mYFAS, GAS, and EDS. CESD = Center for Epidemiologic Studies Depression Scale; COHS = Creature of Habit Scale; EAT = Eating Attitudes Test; EDS = Exercise Dependence Scale; GAS = Gaming Addiction Scale; IAT = Internet Addiction Test; OCI = Obsessive-Compulsive Inventory Revised; LSAS = Liebowitz Social Anxiety Scale; PCLS = Post-traumatic Checklist Scale; PMPUQ = Problematic Mobile Phone Use Questionnaire-Short Version; PSS = Perceived Stress Scale; QABB = Questionnaire About Buying Behavior; SASSV = Smartphone Addiction Scale Short Version; STICSA-T = State-Trait Inventory for Cognitive and Somatic Anxiety: trait; STAI-T = State-Trait Anxiety Inventory: trait; UPPS = UPPS Impulsive Behavior Scale; UPPS-P = UPPS-P Impulsive Behavior Scale; mYFAS = modified Yale Food Addiction Scale 2.0.

and the Exercise Dependence Scale Revised (EDS; 21 items; [57]). Compulsivity was again assessed with the OCI [47].

To assess mental health problems typically associated with problematic reward-seeking and compulsive behaviors, we administered the PSS [49], the PCLS [51], and the CESD [50] as we did in Experiment 1. To target more specific types of anxiety, we also administered the State-Trait Inventory for Cognitive and Somatic Anxiety (STICSA; 42 items; [41]) and the Liebowitz Social Anxiety Scale (LSAS; 24 items; [58]). To assess personality traits related to impulsivity and sensitivity to reward and punishment, we used the UPPS-P Impulsive Behavior Scale (short version; 20 items; [53]) and the Behavioral Inhibition/Activation Systems Scales (BIS/BAS; 24 items; [59]).

Participants completed the questionnaires in two separate days: the COHS, EAT, SASSV, GAS, EDS, PCLS, BIS/BAS, STICSA on day 1 and the UPPS-P, LSAS, OCI, mYFAS, IAT, QABB, PSS, CESD on day 2. The order of the questionnaires was randomized across participants within each day.

For the analysis, we chose to retain only the UPPS-P as a personality questionnaire and to only keep the trait version of the STICSA, which reflects characteristics that are stable over

time. However, all the data are made openly available (see data and code availability statement). Descriptive statistics for the questionnaires included in the analysis for Experiment 2 are reported in Table 2 and the distributions of the questionnaire scores are depicted in Fig 1B. Distributions of the COHS subscales are shown in S1 File.

## Data analysis

Statistical analyses were performed with R [60]. For each experiment, we first applied an exploratory factor analysis (EFA) to the mental health questionnaire subscales to extract common dimensions from the assessed mental health space. We then used network analysis to test whether the two components of habitual behavior were associated with problematic reward-seeking and compulsive behaviors. To further assess the links between habit and specific mental health difficulties, we conducted an additional network analysis at the symptom level on data pooled from Experiments 1 and 2.

**Data preparation** For Experiment 1, we standardized the UPPS and UPPS-P subscales to merge the longer and shorter versions. Given that the positive urgency items were not present in the longer UPPS version, we removed them from the shorter UPPS-P version. In Experiment 2, item 3 of the mYFAS was missing for 45 participants; we imputed the missing data by using the other 12 items as predictors in a logistic regression.

To characterize the level of mental health difficulties present in each sample, we used cutoff scores when available in the literature. These cutoff values were: 19 for the CES-D [50], 20 for the EAT [43], 60 for the LSAS [84], 14 for the OCI [86], 44 for the PCLS [87], 52 for the STAI-T [88], and 43 for the STICSA-T [85]. Moreover, for the GAS, scores of "sometimes" or above on at least half the items were taken as indicators of excessive gaming [56]. A score of 50 or above on the IAT indicated frequent problems due to Internet use [44]. For the mYFAS, the recommended scoring procedure was followed to identify food addiction [54]. A score of 10 or above on the QABB was taken to indicate compulsive buying [83]. A score of 32 or above

**Table 2. Cronbach's $\alpha$, mean and standard deviation of the questionnaires used in the analysis for Experiment 2.**

| Questionnaire | N | $\alpha$ | Mean | SD |
|---|---|---|---|---|
| CESD | 285 | 0.94 | 20.07 | 12.65 |
| COHS | 285 | 0.85 | 86.94 | 14.95 |
| EAT | 285 | 0.89 | 12.11 | 10.95 |
| EDS | 285 | 0.95 | 40.62 | 19.12 |
| GAS | 285 | 0.89 | 11.29 | 5.26 |
| IAT | 285 | 0.91 | 39.81 | 14.32 |
| LSAS | 285 | 0.96 | 59.44 | 27.67 |
| OCI | 285 | 0.89 | 18.68 | 11.43 |
| PCLS | 285 | 0.95 | 39.38 | 16.89 |
| PSS | 285 | 0.89 | 29.26 | 7.63 |
| QABB | 285 | 0.82 | 4.38 | 3.57 |
| SASSV | 285 | 0.87 | 28.42 | 9.82 |
| STICSA-T | 285 | 0.93 | 40.27 | 12.03 |
| UPPS-P | 285 | 0.85 | 44.73 | 8.21 |
| mYFAS | 285 | 0.86 | 1.19 | 1.94 |

$\alpha$ = Chronbach's $\alpha$, SD = standard deviation. CESD = Center for Epidemiologic Studies Depression Scale; COHS = Creature of Habit Scale; EAT = Eating Attitudes Test; EDS = Exercise Dependence Scale; GAS = Game Addiction Scale; IAT = Internet Addiction Test; OCI = Obsessive-Compulsive Inventory Revised; LSAS = Liebowitz Social Anxiety Scale; PCLS = Post-traumatic Checklist Scale; PSS = Perceived Stress Scale; QABB = Questionnaire About Buying Behavior; SASSV = Smartphone Addiction Scale Short Version; STICSA-T = State-Trait Inventory for Cognitive and Somatic Anxiety: trait; UPPS-P = UPPS-P Impulsive Behavior Scale; mYFAS = modified Yale Food Addiction Scale 2.0.

on the SASSV was taken to indicate problematic mobile phone use [55]. For the EDS, the recommended flowchart procedure was followed to identify participants considered to be at risk for exercise dependence [89]. To the best of our knowledge, there are no cutoff scores for the PSS or the UPPS. Fig 1 shows what proportion of participants is considered to have clinically relevant levels of mental health difficulties, and suggests that this was variable within each sample: the proportion of scores above cutoff values was especially high for questionnaires measuring depression (CES-D), anxiety (STAI-T, LSAS, STICSA-T), obsessive-compulsive behaviors (OCI), and problematic smartphone use (SASSV). Conversely, the proportion of problematic eating (EAT, mYFAS) and problematic exercise (EDS) was relatively low.

**Exploratory factor analysis** We used the *psych* package [61] to perform an EFA on the questionnaire subscales assessing different symptoms of mental health problems, which were highly correlated (Figs 2 and 3), using maximum likelihood estimation and an oblimin rotation. To determine the number of factors, we used the following techniques: "parallel analysis" and "minimum average partial procedure" with the *psych* package, "optimal coordinate" and "acceleration factor" with the *nFactors* package [62], and "comparison data" [63] with

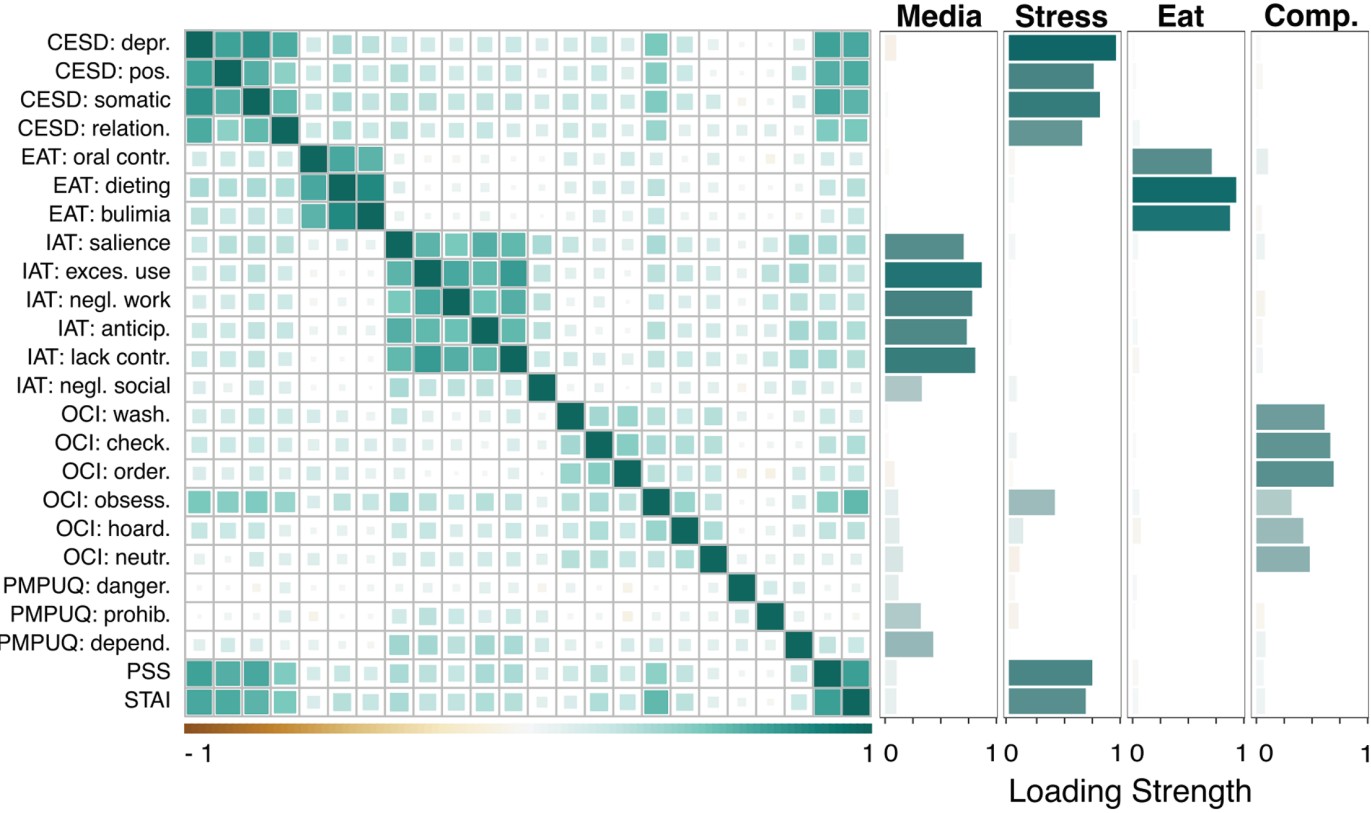

**Fig 2. Correlation matrix and standardized loadings of the subscales used in Experiment 1.** The exploratory factor analysis suggested a four-factor solution: factors were labeled "problematic media consumption" (media), "stress", "problematic eating" (eat), and "compulsivity" (comp.). Standardized loadings for questionnaire subscales into each factor are displayed in green for positive values and yellow for negative values. CESD = Center for Epidemiologic Studies Depression Scale; EAT = Eating Attitudes Test; IAT = Internet Addiction Test; OCI = Obsessive-Compulsive Inventory Revised; PSS = Perceived Stress Scale; PMPUQ = Problematic Mobile Phone Use Questionnaire-Short Version; STAI-T = State-Trait Anxiety Inventory: trait; depr. = depressive affect; pos. = positive mood; relation. = disturbed interpersonal relationships; contr. = control; exces. use = excessive use; negl. = neglect; anticip. = anticipation; wash. = washing; check. = checking; order. = ordering; obsess. = obsessing; hoard. = hoarding; neutr. = neutralizing; danger. = dangerous use; prohib. = prohibited use; depend. = dependent use. N = 381.

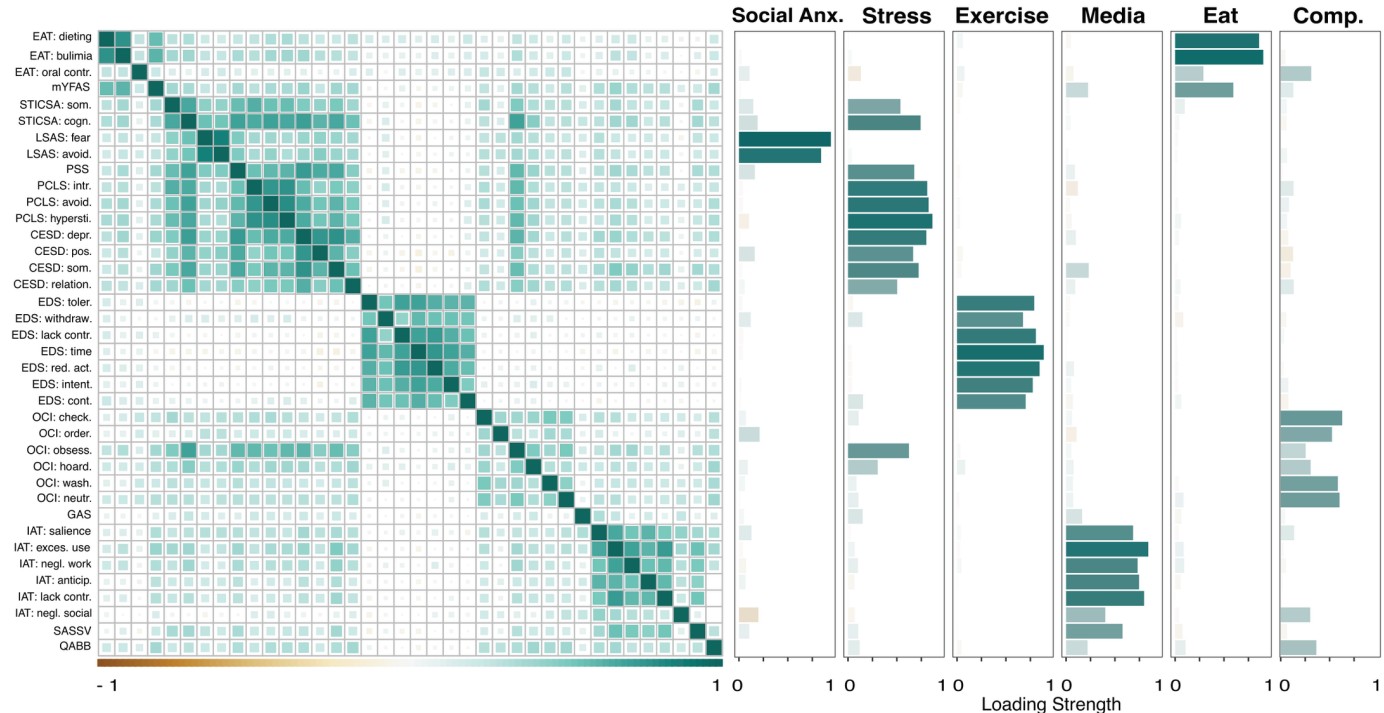

**Fig 3. Correlation matrix and standardized loadings of the subscales used in Experiment 2.** The exploratory factor analysis suggested a six-factor solution: factors were labeled "social anxiety" (social anx.), "problematic exercise" (exercise), "problematic media consumption" (media), "stress", "problematic eating" (eat), and "compulsivity" (comp.). Standardized loadings for questionnaire subscales into each factor are displayed in green for positive values and yellow for negative values. CESD = Center for Epidemiologic Studies Depression Scale; EAT = Eating Attitudes Test; EDS = Exercise Dependence Scale; GAS = Gaming Addiction Scale; IAT = Internet Addiction Test; OCI = Obsessive-Compulsive Inventory Revised; LSAS = Liebowitz Social Anxiety Scale; PCLS = Post-traumatic Checklist Scale; PSS = Perceived Stress Scale; QABB = Questionnaire About Buying Behavior; SASSV = Smartphone Addiction Scale Short Version; STICSA-T = State-Trait Inventory for Cognitive and Somatic Anxiety: trait; mYFAS = modified Yale Food Addiction Scale 2.0. contr. = control; som. = somatic; cogn. = cognitive; avoid. = avoidance; intr. = intrusions; hyperstim. = hyperstimulation; depr. = depressive affect; pos. = positive mood; relation. = disturbed interpersonal relationships; toler. = tolerance; withdraw. = withdrawal; red. act. = reductions in other activities; intent. = intention effects; cont. = continuance; check. = checking; order. = ordering; obsess. = obsessing; hoard. = hoarding; wash. = washing; neutr. = neutralizing; exces. use = excessive use; negl. = neglect; anticip. = anticipation. N = 285.

the *RGenData* package [64]. Assumptions were verified through Bartlett's test of sphericity to ensure that correlations between items were sufficiently large.

**Network analysis** We then entered the extracted mental health factors and the personality questionnaires measuring habit components (COHS automaticity and routine) and impulsivity (UPPS and UPPS-P) into a network analysis using a regularized partial correlation network approach. Partial correlation networks model unique interactions between variables, allowing for the assessment of relationships between variables of interest while taking all other variables into account; in other words, partial correlation networks map out linear prediction and multicollinearity among all variables included in the network [40]. Because it penalizes model complexity, regularization allows for model selection to be performed jointly to parameter estimation, resulting in parsimonious networks which are unlikely to contain spurious edges [40]. This analysis therefore allowed us to parsimoniously model and visualize the complex relationships between habit components and compulsive and problematic reward-seeking behaviors, while controlling for the influence of other variables liable to affect these relationships, such as affective stress and impulsivity.

We estimated a standardized Gaussian Graphical Model with graphical LASSO (Least Absolute Shrinkage and Selection Operator) as a regularization method with the *bootnet* package [65]. The LASSO tuning parameter was selected by minimizing the Extended Bayesian Information Criterion with $\gamma$ = 0.5. The same package was used to estimate the centrality stability coefficient. We used the *qgraph* package [66] to visualize the network, as well as to estimate the edge weights and the node centrality indices [40].

A bootstrap procedure was used to estimate edge weight stability ($n_{boots}$ = 10000; $n_{cases}$ = 50). To estimate the precision of the edges, we report 95% confidence intervals (CIs) reflecting 2.5% and 97.5% quantiles of the bootstrapped sampling distribution of nonzero estimates and the proportion of times the edge was set to zero (prop0).

We used correlation stability coefficients of $r = 0.7$ ($CS_{r=0.7}$) to measure the stability of centrality indices and of edge estimations. $CS_{r=0.7}$ indicates the percentage of the sample that can be dropped while maintaining a correlation of at least $r = 0.7$ between the sample's indices and the bootstrapped indices with a 95% CI.

To further investigate interactions between mental health problems and habit dimensions at the symptom level, we conducted a complementary network analysis with the individual items of questionnaires used in both Experiments 1 and 2 (i.e., the CESD, COHS, EAT, IAT, OCI, and PSS), allowing for a finer-grained examination of how the two components of habitual behavior relate to the symptom space around compulsive and problematic reward-seeking behaviors. To do so, we first ran an exploratory graph analysis (EGA) with a Walktrap algorithm [67] on the individual items of the five questionnaires assessing mental health problems to identify underlying communities. We then entered the COHS subscales into a network with all the individual items assessing symptoms.

## Results

### Experiment 1

**Factor analysis**   To reduce the dimensionality of the complex mental health space considered, an EFA was performed on the 24 questionnaire subscales (Fig 2). The correlations between subscales were sufficiently large for an EFA ($X^2_{(276)}$ = 4362.016, $p < .001$; Fig 2). The factor analyses we applied converged toward a four-factor solution (Fig 2).

The first factor was characterized by high loadings from subscales assessing problematic use of the Internet and mobile phone (IAT: salience, IAT: excessive use, IAT: neglect work, IAT: lack of control, IAT: neglect social life, IAT: anticipation, PMPUQ: prohibited use, PMPUQ: dependent use); we consequently labeled this factor "problematic media consumption". The second factor was characterized by high loadings from subscales related to stress (PSS), depression (CESD: depressed affect, CESD: positive affect, CESD: somatic complaints, CESD: disturbed interpersonal relationships), and anxiety (STAI-T and OCI: obsessing); we consequently labeled this factor "stress". Subscales assessing problematic eating (EAT: oral control, EAT: dieting, EAT: bulimia) loaded on a third factor, which we labeled "problematic eating". Finally, the fourth factor was characterized by high loadings from subscales associated with compulsive behaviors (OCI: washing, OCI: checking, OCI: ordering, OCI: hoarding, OCI: neutralizing); we consequently labeled this factor "compulsivity".

The validity coefficients ($R^2$ = 0.951, 0.965, 0.963, 0.877) assessing the potential impact of factor score indeterminacy [68] were satisfactory, allowing factor scores derived from the EFA to be used in a network analysis (see [69] for a similar approach).

**Network analysis**   Fig 4B illustrates the dynamic network we used to estimate the connections between the two components of habitual behavior, impulsivity, and the EFA-extracted

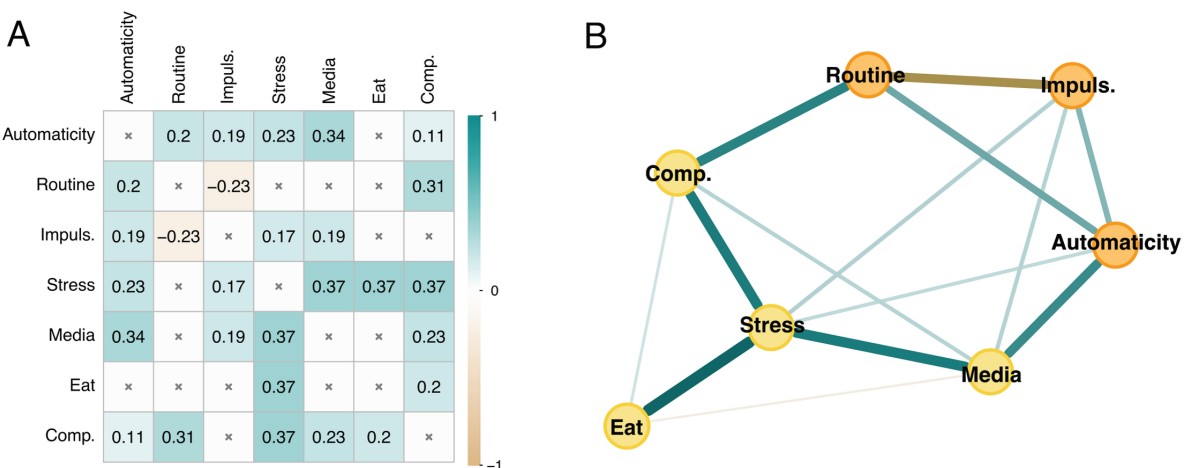

**Fig 4. Experiment 1 network.** A: Correlation matrix of dimensions used as nodes in the network. Only correlations with $p < 0.05$ are displayed. B: Network of relationships between impulsivity and habit components (orange) and mental health factors obtained from the exploratory factor analysis (yellow). Green edges represent positive connections, yellow edges represent negative connections; thicker edges represent stronger connections. Comp = compulsivity; Eat = problematic eating; Impuls. = impulsivity; Media = problematic media consumption. $N = 381$.

mental health factors. These are represented as nodes in the network, and their connections are represented as edge weights. The edge weights can be interpreted as the correlation between two nodes while controlling for all other variables in the network (i.e., partial correlation; [40]). We also computed the centrality index of each node by assessing its expected influence.

Stress was the node with the strongest centrality (Table 3). It had a strong and direct influence on all problematic behavior nodes: problematic eating ($r_p = 0.291$; bootstrap: *mean* = 0.279, 95% CI = $[0.187, 0.375]$, prop0 = 0.000), problematic media consumption ($r_p = 0.248$; bootstrap: *mean* = 0.244, 95% CI = $[0.162, 0.327]$, prop0 = 0.000), and compulsivity ($r_p = 0.253$; bootstrap: *mean* = 0.248, 95% CI = $[0.156, 0.339]$, prop0 = 0.000). Strikingly, compulsivity and problematic media consumption were differently associated with the two habitual behavior components: compulsivity was directly connected to routine ($r_p = 0.242$; bootstrap: *mean* = 0.230, 95% CI = $[0.135, 0.323]$, prop0 = 0.000), whereas problematic media consumption was directly connected to automaticity ($r_p = 0.234$; bootstrap: *mean* = 0.225, 95% CI = $[0.133, 0.314]$, prop0 = 0.0001). Similarly to what has been previously observed with the original [21] and the German [25] versions of the COHS, impulsivity was negatively associated with routine ($r_p = -0.221$; bootstrap: *mean* = -0.201, 95% CI = $[-0.308, -0.091]$, prop0 = 0.001) and positively associated with automaticity ($r_p = 0.136$; bootstrap: *mean* = 0.123, 95% CI = $[0.035, 0.229]$, prop0 = 0.033). Impulsivity was also directly associated with problematic media consumption ($r_p = 0.085$; bootstrap: *mean* = 0.084, 95% CI = $[0.018, 0.186]$, prop0 = 0.087). In line with previous research [25], the two components of habitual behavior were positively connected ($r_p = 0.168$; bootstrap: *mean* = 0.151, 95% CI = $[0.049, 0.263]$, prop0 = 0.016).

We estimated the stability of the dynamic network through correlation stability coefficients, which indicated a satisfactory stability for edges ($CS_{r=0.7} = 0.522$) and expected influence ($CS_{r=0.7} = 0.535$).

**Table 3. Expected influence of the graphical LASSO network nodes.**

|  | Experiment 1 | Experiment 2 |
|---|---|---|
| Automaticity | 0.405 | 0.597 |
| Routine | -0.989 | -0.372 |
| Problematic media | 0.438 | 0.255 |
| Stress | 1.539 | 1.997 |
| Problematic eating | -0.557 | -0.241 |
| Compulsivity | 0.488 | -0.602 |
| Impulsivity | -1.323 | 0.068 |
| Social anxiety | - | 0.037 |
| Problematic exercise | - | -1.740 |

## Experiment 2

**Factor analysis** Similarly to Experiment 1, an EFA was performed on the 38 questionnaire subscales to reduce the dimensionality of the complex mental health space considered (Fig 3). The correlations between subscales were sufficiently large for an EFA ($X^2_{(703)}$ = 6930.254, $p < .001$; Fig 3). The factor extraction analyses that we applied converged toward a six-factor solution (Fig 3).

The first factor was characterized by high loadings from subscales assessing social anxiety (LSAS: fear, LSAS: avoidance); we consequently labeled this factor "social anxiety". The second factor was characterized by high loadings from subscales related to stress (PCLS: intrusions, PCLS: avoidance, PCLS: hyperstimulation, PSS), anxiety (STICSA-T: somatic, STICSA-T: cognitive, OCI: obsessing), and depression (CESD: depressed affect, CESD: positive affect, CESD: somatic complaints, CESD: disturbed interpersonal relationships); we consequently labeled this factor "stress". Subscales measuring problematic exercise (EDS) loaded on a third factor, which we labeled "problematic exercise". The fourth factor was characterized by high loadings from subscales related to problematic use of the Internet (IAT: salience, IAT: excessive use, IAT: neglect work, IAT: lack of control, IAT: neglect social life, IAT: anticipation) and of the mobile phone (SASSV); we consequently labeled this factor "problematic media consumption". Subscales assessing problematic eating (EAT: dieting, EAT: bulimia, EAT: oral control, mYFAS) loaded on a fifth factor, which we labeled "problematic eating". Finally, the last factor was characterized by high loadings from subscales assessing compulsive behaviors (OCI: checking, OCI: hoarding, OCI: washing, OCI: neutralizing, QABB); we consequently labeled this factor "compulsivity".

As in Experiment 1, the validity coefficients ($R^2$ = 0.968, 0.973, 0.965, 0.955, 0.949, 0.888) assessing the potential impact of factor score indeterminacy [68] were satisfactory, allowing factor scores derived from the EFA to be used in a subsequent network analysis.

**Network analysis** Fig 5 illustrates the dynamic network estimating the connections between the two components of habitual behavior, impulsivity, and the EFA-extracted mental health factors. These are represented as nodes in the network, and their connections are represented as edge weights. We computed the centrality index of each node by assessing its expected influence.

Stress was again the node with the strongest centrality (Table 2). It had a strong and direct influence on all problematic behavior nodes, except for problematic exercise: problematic eating ($r_p$ = 0.270; bootstrap: *mean* = 0.248, 95% CI = [0.139, 0.349], prop0 = 0.000), social anxiety ($r_p$ = 0.329; bootstrap: *mean* = 0.317, 95% CI = [0.212, 0.416], prop0 = 0.000), problematic media consumption ($r_p$ = 0.182; bootstrap: *mean* = 0.171, 95% CI = [0.066, 0.277], prop0 = 0.001), and compulsivity ($r_p$ = 0.136; bootstrap: *mean* = 0.124, 95% CI = [0.028, 0.231], prop0

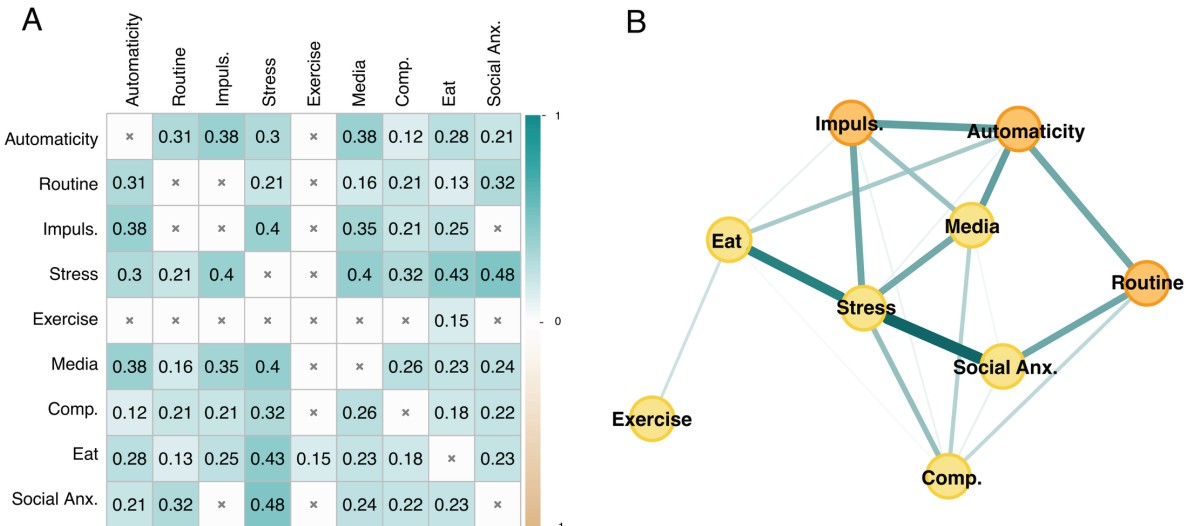

**Fig 5. Experiment 2 network.** A: Correlation matrix of dimensions used as nodes in the network. Only correlations with *p* < 0.05 are displayed. B: Network of relationships between impulsivity and habit components (orange) and mental health factors obtained from the exploratory factor analysis (yellow). Green edges represent positive connections, yellow edges represent negative connections; thicker edges represent stronger connections. Anx. = anxiety; Exercise = problematic exercise; Media = problematic media consumption; Eat = problematic eating; Comp. = compulsivity. *N* = 285.

= 0.017). Like in Experiment 1, the two habitual behavior components were differently related to mental health factors: routine was directly connected to compulsivity ($r_p$ = 0.084; bootstrap: *mean* = 0.081, 95% CI = [0.016, 0.201], prop0 = 0.153), whereas automaticity was directly connected to problematic media consumption ($r_p$ = 0.204; bootstrap: *mean* = 0.193, 95% CI = [0.089, 0.295], prop0 = 0.0005) and problematic eating ($r_p$ = 0.110; bootstrap: *mean* = 0.096, 95% CI = [0.000, 0.199], prop0 = 0.060). Similarly, impulsivity was positively associated with automaticity ($r_p$ = 0.204; bootstrap: *mean* = 0.207, 95% CI = [0.098, 0.317], prop0 = 0.0004) and problematic media consumption ($r_p$ = 0.130; bootstrap: *mean* = 0.128, 95% CI = [0.029, 0.234], prop0 = 0.014). However, we did not replicate the negative association between impulsivity and routine. Aligning with existing evidence [25] and Experiment 1, the two components of habitual behavior were positively connected ($r_p$ = 0.184; bootstrap: *mean* = 0.181, 95% CI = [0.069, 0.296], prop0 = 0.004). Social anxiety was also positively connected with routine ($r_p$ = 0.187; bootstrap: *mean* = 0.174, 95% CI = [0.072, 0.273], prop0 = 0.002) but not with automaticity. Problematic exercise was only associated—positively—with problematic eating ($r_p$ = 0.064; bootstrap: *mean* = 0.073, 95% CI = [0.018, 0.219], prop0 = 0.258).

The correlation stability coefficients indicated a satisfactory stability for edges ($CS_{r=0.7}$ = 0.463) and expected influence ($CS_{r=0.7}$ = 0.477).

## Symptoms network analysis

The EGA conducted on the individual items of the five questionnaires assessing mental health problems revealed 4 communities: a community for the OCI, EAT, and IAT each, and one community covering items from both the CESD and the PSS.

Fig 6 illustrates the network estimating the connections between the individual items assessing symptoms and the two COHS subscales. Routine was most strongly connected with items assessing compulsive symptoms such as ordering (item 15; $r_p$ = 0.096; bootstrap:

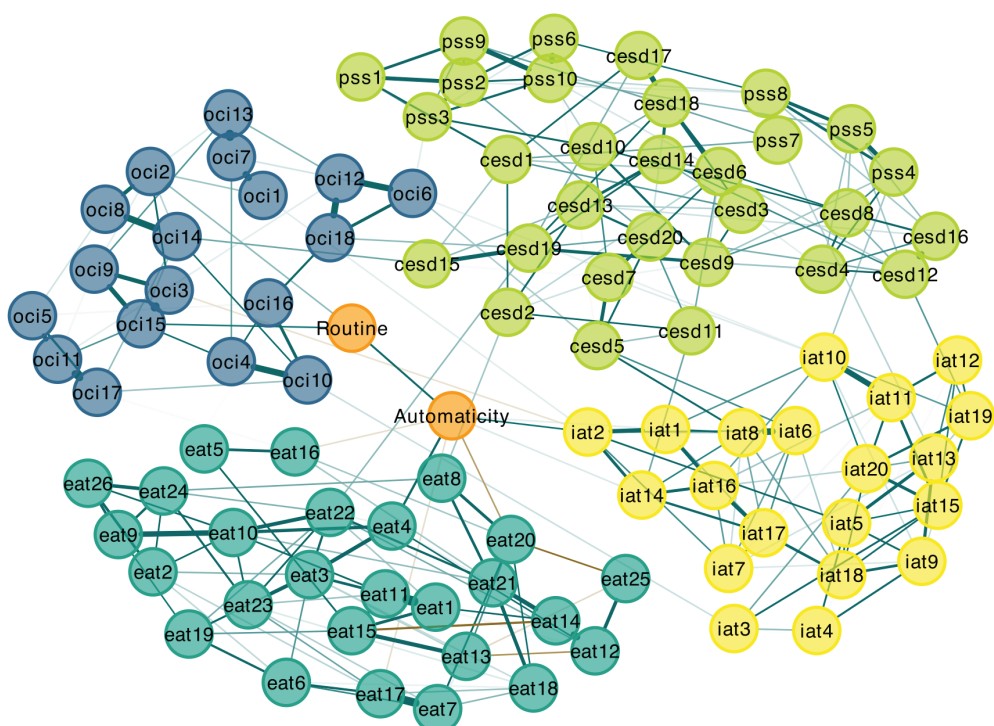

**Fig 6. Network of relationships between symptoms and habitual behavior components.** Green edges represent positive connections and yellow edges represent negative connections; thicker edges represent stronger connections. Color of item nodes represents communities identified by the Walktrap algorithm. cesd = Center for Epidemiologic Studies Depression Scale items (light green); eat = Eating Attitudes Test items (dark green); oci = Obsessive-Compulsive Inventory Revised items (blue); iat = Internet Addiction Test items (yellow); pss = Perceived Stress Scale items (light green). Only edges larger than 0.05 are depicted. $N$ = 666.

$mean$ = 0.085, 95% CI = $[0.025, 0.144]$, prop0 = 0.006) and checking (item 2; $r_p$ = 0.074; bootstrap: $mean$ = 0.062, 95% CI = $[0.011, 0.127]$, prop0 = 0.037). By contrast, automaticity was most strongly connected with symptoms from different communities such as problematic use of the Internet (item 2; excessive use, inability to control time spent online: $r_p$ = 0.099; bootstrap: $mean$ = 0.089, 95% CI = $[0.032, 0.148]$, prop0 = 0.004) and problematic eating (item 4; episodes of excessive eating, inability to stop eating: $r_p$ = 0.137; bootstrap: $mean$ = 0.117, 95% CI = $[0.062, 0.173]$, prop0 = 0.000). Although they belong to different communities, these two items are characterized by an inability to control reward consumption. Interestingly, automaticity played different roles within the problematic eating community: it was positively associated with the only item assessing episodes of excessive eating (item 4: $r_p$ = 0.137; bootstrap: $mean$ = 0.117, 95% CI = $[0.062, 0.173]$, prop0 = 0.000) and negatively associated with items assessing dieting (item 17: $r_p$ = −0.063; bootstrap: $mean$ = −0.044, 95% CI = $[−0.115, −0.004]$, prop0 = 0.140; item 16: $r_p$ = −0.061; bootstrap: $mean$ = −0.041, 95% CI = $[−0.112, −0.004]$, prop0 = 0.163) and oral control (item 20 $r_p$ = −0.087; bootstrap: $mean$ = −0.072, 95% CI = $[−0.142, −0.009]$, prop0 = 0.024).

## Discussion

Habit has been implicated in a variety of mental health problems. In this study, we aimed to investigate whether taking the multidimensionality of habit into account could provide

insight into the links between habit and associated mental health problems. Specifically, we examined how two components of habit—automaticity and routine—relate to compulsive and problematic reward-seeking behaviors.

Across two experiments, we showed that both compulsive and problematic reward-seeking behaviors were related to habit, extending existing evidence for the implication of a shared dimension related to habitual processes across these mental health difficulties [9,72]. The experiments also showed, however, that two components of habitual behavior were differentially related to mental health problems: routine was associated with compulsivity and social anxiety, while automaticity was associated with problematic media consumption and problematic eating. These findings suggest that the distinctions between automaticity and routine might help better understand how an imbalance between goal-directed and habitual control can give rise to different problematic behaviors.

A possible explanation is that the intervention of habitual processes at different levels of behavior, from its instigation to its performance, can render individuals vulnerable to different problematic behaviors. Automaticity captures the tendency to experience bouts of purely habitually-driven behavior, in which neither behavioral initiation nor execution are under goal-directed control [20]. Routine instead denotes a preference for structure or behavioral regularity, suggesting that habitual processes may be deployed in the context of goal-directed behavior, for instance by facilitating its execution [20].

The tendency to have behavior hijacked by habit, as measured by automaticity, could make individuals particularly prone to engage in reward-seeking behaviors when encountering stimuli which have been associated with rewards. The western, highly digitalized environment in which the present study took place is saturated with cues for both media and food rewards [54,70], presenting countless opportunities for habit to drive behavior toward reward pursuit in individuals with high automaticity, despite goal-directed opposition. These episodes of severe imbalance between goal-directed and habitual control are further evidenced by the symptoms network analysis (Fig 6), which showed that automaticity was specifically associated with symptoms related to craving and self-control failures such as excessive eating or Internet use, and negatively connected to symptoms related to the implementation of rigid dysfunctional behaviors such as dieting. From this perspective, the association between automaticity and different forms of problematic reward-seeking behavior is congruent with evidence showing that substance use disorder is specifically associated with automaticity [22,23]. Indeed, problematic media consumption is considered to be an increasingly widespread problematic reward-seeking behavior which shares characteristics with addiction, such as psychological dependence and reduced self-regulation [70,71]. Similarly, the association between automaticity and problematic eating behavior could result from an increased difficulty in resisting cue-induced food consumption. The fact that this association was found in Experiment 2 but not Experiment 1 could be explained by the inclusion of the mYFAS scale—a tool specifically geared toward the identification of addictive-like eating behavior—or by the higher prevalence of problematic eating—as measured with the EAT scale. Further analyses conducted on Experiment 2 data without the mYFAS still showed an association between automaticity and problematic eating (S2 File), thereby suggesting that an explanation in terms of sample levels of problematic eating may be more likely.

The association between routine and compulsivity, conversely, illustrates how a tendency to pursue structure and repetition in daily life could, through an increased reliance on habitual execution, be a contributing factor to the generation of situations in which goal-directed intentions are dysfunctionally diverted by habit. Individuals who frequently engage in rigid behavioral sequences (e.g., always following a set of steps in a specific order before going out) may effectively practice these action chunks to the point where their execution becomes

strongly habitual [91]. This repetition may favor the formation of behavioral loops in which individuals remain stuck, despite goal-directed action selection (e.g., checking that the stove is turned off or that the door is locked with the goal of avoiding harm) [26]. This could explain why routine was found to be most strongly associated with rigid dysfunctional behaviors related to checking and ordering in the symptoms network analysis (Fig 6).

While compulsivity has previously been found to be positively associated with both automaticity and routine, with a stronger link to routine [24,25], we only found evidence for an association between compulsivity and routine. This may be due to the inclusion of additional nodes in our networks, such as stress, which could be mediating the effect. Indeed, we did find small correlations between compulsivity and automaticity in the descriptive correlational matrices (Figs 4A and 5A).

The role of impulsivity in the network also highlights how a predominance of habitual over goal-directed control can interact with other psychological traits, resulting in different behavioral manifestations. While impulsivity had a positive association with automaticity and problematic media consumption in both experiments, it had a strong negative association with routine in Experiment 1 and no association with routine in Experiment 2. These results are mostly in line with previous findings showing a positive link between impulsivity and automaticity and a negative link between impulsivity and routine [24,25]. These differential associations may relate to the different degree of goal-directed control involved in automatic and routine behaviors. Indeed, facets of impulsivity such as urgency or lack of premeditation share important characteristics with habitual behavior [73]. Individuals with high levels of impulsivity may be more prone to engage in automatic behaviors, which are fully under habitual control, while individuals with lower levels of impulsivity may more readily engage and persist in routine behaviors, which require goal-directed involvement to be initiated or executed.

Our findings provide evidence for a central role of stress in various types of problematic behaviors: stress had a positive link with problematic eating, problematic media consumption, compulsive behaviors, and social anxiety. This is congruent with literature showing that stress exacerbates symptomatology in a variety of mental health problems such as addiction [29,30], binge eating disorder [74], and obsessive-compulsive disorder [75]. Stress has also been found to affect the balance between habitual and goal-directed processes [10,31,33,34]: specifically, stress appears to accelerate the shift from goal-directed to habitual behavior that is hypothesized to naturally occur with repetition in a specific context [33,76,77]. Interestingly, despite the fact that stress has been shown to modulate both problematic behaviors and the development of habits, stress exerted a direct influence on all problematic behavior nodes—except for problematic exercise—with no evidence of mediation by habit. This could be due to stress acting on problematic behaviors through mechanisms separate from habit, but could also be explained by specificities of our study not allowing us to capture more complex interactions in terms of mechanisms.

Contrary to our expectations, problematic exercise emerged as an isolated node in our network, and was only associated with problematic eating. We expected problematic exercise to be associated with habit components and other problematic reward-seeking behaviors, reflecting a common dimension. This result could suggest that mechanisms involved in problematic exercise are distinct from those involved in other reward-seeking or habitual behavior. Alternatively, the absence of that link in the present study might be related to the low levels of problematic exercise in our sample, which could have prevented us from reliably identifying the relationships between problematic exercise, various kinds of reward-seeking behaviors, and habits.

This study has several limitations. First, participants were recruited from a non-clinical population. A growing number of studies taking a dimensional perspective on mental health have identified transdiagnostic factors in community samples (e.g., [72,90]); nevertheless, whether the associations we identified in this study between habit and problematic behaviors interact with severity levels remains an open question. In addition, we did not ask participants whether they had received a mental health diagnosis, or whether they were undergoing any type of treatment for mental health difficulties. We therefore do not know whether the relatively high proportion of clinically relevant levels of depression, anxiety, obsessive-compulsive behaviors, and mobile phone use was associated with formal diagnoses and treatment, and whether this impacted our findings. Relatedly, our sample was imbalanced in terms of gender, which may limit the generalizability of our findings given that gender plays an important role in mental health [92]. The higher proportion of women in our sample may have influenced the prevalence of various mental health difficulties, which may in turn have impacted the resulting networks. Gender differences could also exist in the balance between goal-directed and habitual behavior; alternatively, gender may moderate the link between habitual processes and different mental health difficulties. Further research is needed to disentangle these effects. Finally, because we only used questionnaires, we cannot formulate causal or mechanistic explanations for the relationships we identified between components of habitual behavior and the mental health problems we measured. Designing experimental paradigms which allow different habit components to be measured separately will be crucial to continue to characterize the links between habits and mental health.

Overall, our results underline how considering the non-unitary architecture of habits may allow for a more granular understanding of the role of habit in mental health, showing that different components of habitual behavior are differentially associated with compulsive and problematic reward-seeking behaviors. This perspective could also contribute to addressing the difficulty of experimentally studying habits in humans [33,78–80].

## Supporting information

**S1 File. French validation of the Creature of Habit Scale.** Psychometric properties of the French Creature of Habit Scale.
(PDF)

**S2 File. Replication.** Additional factor and network analyses on the restricted set of questionnaires used in both Experiments 1 and 2.
(PDF)

## Acknowledgments

The authors would like to thank Tamara Corino, Maelys Denis-Bonnin, and Anaïs Putti García for their assistance in data collection and Dr. Ben Meuleman for his thoughtful advice on statistical analysis.

## Author contributions

**Conceptualization:** Lavinia Wuensch, Yoann Stussi, David Sander, Julie Péron, Eva R. Pool.

**Data curation:** Lavinia Wuensch, Théo Vernede, Eva R. Pool.

**Formal analysis:** Lavinia Wuensch, Théo Vernede, Eva R. Pool.

**Funding acquisition:** Eva R. Pool.

**Methodology:** Lavinia Wuensch, Ryan J. Murray, Eva R. Pool.

**Supervision:** Eva R. Pool.

**Visualization:** Eva R. Pool.

**Writing – original draft:** Lavinia Wuensch, Eva R. Pool.

**Writing – review & editing:** Yoann Stussi, Ryan J. Murray, David Sander, Julie Péron.

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
