## [Decision Letter · Decision Letter 0]

29 Dec 2024

PMEN-D-24-00495

Differential influence of habit components on compulsive and problematic reward-seeking behavior

PLOS Mental Health

Dear Dr. Wuensch,

Thank you for submitting your manuscript to PLOS Mental Health. After careful consideration, we feel that it has merit but does not fully meet PLOS Mental Health’s publication criteria as it currently stands.

As you will see from the reviewers' comments, the manuscript will benefit from more detailed descriptions of the sample characteristics, more explicit definition of conceptualisation, and justification of analytical approaches. Better links between the two studies would also be appreciated. 

Therefore, we invite you to submit a revised version of the manuscript that addresses the points raised during the review process.

Please submit your revised manuscript by Feb 12 2025 11:59PM.

Note that this is the default revision deadline. But if you will need more time than this to complete your revisions, please reply to this message or contact the journal office at mentalhealth@plos.org. Please include the following items when submitting your revised manuscript:

We look forward to receiving your revised manuscript.

Kind regards,

Dr. Lei Zhang

Academic Editor

PLOS Mental Health

1. We ask that a manuscript source file is provided at Revision. Please upload your manuscript file as a .doc, .docx, .rtf or .tex.

2. Please provide separate figure files in .tif or .eps format.

For more information about figure files please see our guidelines: https://journals.plos.org/mentalhealth/s/figures 

https://journals.plos.org/mentalhealth/s/figures#loc-file-requirements 

3. Please provide an Author Summary. This should appear in your manuscript between the Abstract (if applicable) and the Introduction, and should be 150–200 words long. The aim should be to make your findings accessible to a wide audience that includes both scientists and non-scientists. Sample summaries can be found on our website under Submission Guidelines:

https://journals.plos.org/globalpublichealth/s/submission-guidelines#loc-parts-of-a-submission.

Reviewers' comments:

Reviewer's Responses to Questions

**Comments to the Author**

1. Does this manuscript meet PLOS Mental Health’s publication criteria? Is the manuscript technically sound, and do the data support the conclusions? The manuscript must describe methodologically and ethically rigorous research with conclusions that are appropriately drawn based on the data presented.

Reviewer #1: Yes

Reviewer #2: Partly

2. Has the statistical analysis been performed appropriately and rigorously?

Reviewer #1: I don't know

Reviewer #2: Yes

3. Have the authors made all data underlying the findings in their manuscript fully available (please refer to the Data Availability Statement at the start of the manuscript PDF file)?

Reviewer #1: Yes

Reviewer #2: Yes

4. Is the manuscript presented in an intelligible fashion and written in standard English?

Reviewer #1: Yes

Reviewer #2: Yes

5. Review Comments to the Author

Reviewer #1: In this study, the authors explored the roles of different components of habitual behavior—namely, routine and automaticity—in relation to mental health problems. Using questionnaires completed by 666 participants, they conducted dynamic network analyses that showed routine is associated with compulsivity, while automaticity, in contrast, is linked to problematic media consumption. While previous studies demonstrating habit as a unitary variable is associated with compulsivity (Everitt & Robbins, 2005; Gillian & Robbins, 2014), this conclusion this conclusion opens up a new dimension for the study of habits.

The strength of this paper lies in its clear structure and rigorous application of analytical methods. Although I am not an expert in questionnaire analysis, I found the authors’ explanation of their methodology both easy to follow and convincingly thorough. The findings regarding the associations between routine, automaticity, compulsivity, and reward-seeking behaviors were reliably reproduced across two experiments, despite using some different questionnaires. There are inconsistencies between the two experiments, but they are reasonably justified in the discussion section. Therefore, the study is quite complete. The primary weakness of the paper lies in its limited illustration of how the key findings—specifically, the associations between routine, automaticity, compulsivity, and reward-seeking behaviors—advance the current research landscape. The following discussion will focus on how the authors could better articulate the broader contributions of these findings to the research community.

Major issues:

One factor that limits the impact of the conclusion is that the authors did not clearly differentiate between routine and automaticity. The current descriptions of these two components, found in lines 45-55, do not provide a clear distinction between them. It wasn’t until I referred to the cited work “Habits” by Robbins & Costa (2017) that I grasped their nuance differences. As mentioned in the discussion (lines 444-449), one of the goals of this study is to inspire the development of paradigms for experimentally investigating the observed associations. To achieve this, the authors could benefit from providing more precise descriptions, such as specific behavioral patterns driven by each component or concrete examples from daily life. This clarity would help experimenters better define and operationalize the two components in future research and this can help people better see the study’s contribution.

A second area where the authors could improve is by discussing in greater depth how their conclusions can “help to better understand the role of habit in mental health.” For example, in line 388, the authors mention that the strong association between automaticity and problematic media consumption implies a link to addiction. It would be helpful if the authors further explored the relationship between routine and mental disorders associated with compulsivity, such as OCD, as well as the relationship between automaticity and addiction. Expanding on these points could provide valuable insights for improving the diagnosis and understanding of mental health conditions.

Minor issues:

I have noticed that the sample is quite imbalanced in terms of gender, with 506 female participants, 169 male participants, and 6 non-binary participants (line 461). Given that gender is an important factor in understanding mental health, could the authors discuss any potential gender-specific impacts on the two target associations? How might these differences influence the findings related to routine, automaticity, compulsivity, and reward-seeking behaviors?

Summary:

Overall, the work is thorough. However, adding more detail about the broader implications of the conclusions could make the paper more appealing to a wider audience, particularly those more interested in the findings than in the methodology.

Reference:

Everitt, B., Robbins, T. Neural systems of reinforcement for drug addiction: from actions to habits to compulsion. Nat Neurosci 8, 1481–1489 (2005).

Gillan, C. M., & Robbins, T. W. (2014). Goal-directed learning and obsessive–compulsive disorder. Philosophical Transactions of the Royal Society B: Biological Sciences, 369(1655), 20130475.

Robbins, T. W., & Costa, R. M. (2017). Habits. Current biology : CB, 27(22), R1200–R1206.

Reviewer #2: In their paper “Differential influence of habit components on compulsive and

problematic reward-seeking behavior”, the authors examine how habitual behavior relates to compulsive and problematic reward-seeking behavior.

This is a well powered study looking for latent factors that can impact common psychiatric symptoms in a range of disorder. I also appreciate that the data and code are publicly available. However, it is important to note that participants were undergraduate students or online study participants, and I am assuming without psychiatric disorders. Therefore, while the study can inform research on transdiagnostic markers of mental health it is primarily a study of individual differences.

Authors claim that they follow a transdiagnostic approach. I see how studying underlying dimensions of problem behaviors is relevant for transdiagnostic research, however, the current study did not include individuals with any diagnoses, so this claim must be toned down. What’s more, authors should provide much more details on their sample in terms of reporting certain mental health problems or diagnoses.

In a similar vein with respect to the measures, how did authors select these specific measures based on the problem behaviors they were interested in? I am familiar with some but not with others. Are they normed in the population they are studying, are there cutoff values for clinical group that would indicate whether someone is in a clinically significant spectrum? These details are currently missing. I would want to see certain cutoff ranges in the histogram plots in order to have a better idea of the sample.

The analyses need to be motivated much more and that goes back to the question of why exactly certain measures were selected in experiment 1 and why the additional instruments were added in experiment 2.

As I understand both EFA and network analyses approaches are highly exploratory and the factor structure and network analyses really depend on the inputs so I am not surprised to see the differences in the factor solutions for experiments 1 and 2. It seems to me that authors had specific hypotheses about how these factors would predict scores on certain questionnaires. Could authors test this with a standard GLM or LME model? I am assuming that the questionnaires also have various items and scales that assess habitual behavior and relationships would reflect this overlap in common measures. So I would want to see with a stepwise approach that latent factors of questionnaire measures, particularly those that are less related to habitual behavior (e.g., media use vs. compulsiveness, which includes having fixed routines) are in fact predicted by habitual behavior.

I would also want to see a replication of the structure with the initial instrument set from experiment 1 to experiment 2.

Finally, the paper is hard to read in this form because figures and figure captions are separated and in methods and results, analyses are not sufficiently motivated. It would be important to improve clarity by motivating the analyses more and tying them back to the hypotheses.

6. PLOS authors have the option to publish the peer review history of their article (what does this mean?). If published, this will include your full peer review and any attached files.

**Do you want your identity to be public for this peer review?** For information about this choice, including consent withdrawal, please see our Privacy Policy.

Reviewer #1: No

Reviewer #2: No

---

## [Decision Letter · Decision Letter 1]

8 Apr 2025

PMEN-D-24-00495R1

Differential influence of habit components on compulsive and problematic reward-seeking behavior

PLOS Mental Health

Dear Dr. Wuensch,

Thank you for submitting your manuscript to PLOS Mental Health. After careful consideration, we feel that it has merit but does yet not fully meet PLOS Mental Health’s publication criteria as it currently stands. Therefore, we invite you to submit a revised version of the manuscript that addresses the points raised during the review process, especially on self-reported diagnoses (if any) or discuss its potential limitation to the current study.

We look forward to receiving your revised manuscript.

Kind regards,

Lei Zhang

Academic Editor

PLOS Mental Health

Journal Requirements:

Additional Editor Comments (if provided):

Reviewers' comments:

Reviewer's Responses to Questions

**Comments to the Author**

1. If the authors have adequately addressed your comments raised in a previous round of review and you feel that this manuscript is now acceptable for publication, you may indicate that here to bypass the “Comments to the Author” section, enter your conflict of interest statement in the “Confidential to Editor” section, and submit your "Accept" recommendation.

Reviewer #1: All comments have been addressed

Reviewer #2: (No Response)

2. Does this manuscript meet PLOS Mental Health’s publication criteria? Is the manuscript technically sound, and do the data support the conclusions? The manuscript must describe methodologically and ethically rigorous research with conclusions that are appropriately drawn based on the data presented.

Reviewer #1: Yes

Reviewer #2: Yes

3. Has the statistical analysis been performed appropriately and rigorously?

Reviewer #1: Yes

Reviewer #2: Yes

4. Have the authors made all data underlying the findings in their manuscript fully available (please refer to the Data Availability Statement at the start of the manuscript PDF file)?

Reviewer #1: (No Response)

Reviewer #2: Yes

5. Is the manuscript presented in an intelligible fashion and written in standard English?

Reviewer #1: Yes

Reviewer #2: Yes

6. Review Comments to the Author

Reviewer #1: I am generally satisfied with the revisions made and the improvements to the manuscript. It is evident that the authors have put considerable effort into clarifying the distinction between routine and automaticity, as well as expanding the implications of their findings. I believe these revisions greatly enhance both the clarity and the contribution of the study.

The authors’ expanded explanation in the introduction now provides a much clearer differentiation between routine and automaticity. The inclusion of real-life examples and descriptions significantly improves the conceptual distinction between these two components, and I believe this will help readers better understand the study’s contribution.

Additionally, the authors have made a strong effort to expand the discussion on the broader implications of their findings. The added details regarding how these habitual components may contribute to mental health conditions such as substance use disorder, and problematic eating provide valuable context and will be helpful for researchers in clinical psychology and related fields.

I have no further concerns regarding the publication of this manuscript in its current form.

Reviewer #2: Authors have much improved the manuscript in their revision. An issue that remains is that authors do not report any mental health self report questions in their sample. If this study was done with a clinical focus in mind, I would think that asking subjects to self report diagnoses, treatments including medication would be important. This is particularly important given that a significant proportion of participants self-report anxiety and OCD symptoms in the clinically significant range. If this has not been assessed and or information is not available to authors, this should be clearly stated as a limitation of the current study

7. PLOS authors have the option to publish the peer review history of their article (what does this mean?). If published, this will include your full peer review and any attached files.

**Do you want your identity to be public for this peer review?** For information about this choice, including consent withdrawal, please see our Privacy Policy.

Reviewer #1: No

Reviewer #2: No

---

## [Editor Report · Decision Letter 2]

23 Apr 2025

Differential influence of habit components on compulsive and problematic reward-seeking behavior

PMEN-D-24-00495R2

Dear Ms Wuensch,

We are pleased to inform you that your manuscript 'Differential influence of habit components on compulsive and problematic reward-seeking behavior' has been provisionally accepted for publication in PLOS Mental Health. Congratulations!

Best regards,

Lei Zhang

Academic Editor

PLOS Mental Health